# Impact of Graduate Student Expansion and Innovative Human Capital on Green Total Factor Productivity

**Hao Yao** [1] , **Xiulin Gu** [2,*] **and Qing Yu** [2,*]

1   Institute of Higher Education, Tongji University, Shanghai 200092, China
2   School of Education, Soochow University, Suzhou 215123, China
*   Correspondence: xlgu@suda.edu.cn (X.G.); yuqing@suda.edu.cn (Q.Y.)

**Abstract:** Using data from 30 provinces and cities in China from 2005–2018, panel regression models, mediation models, quantile regressions and threshold regressions were used to examine the relationship between graduate student size expansion, innovative human capital and green total factor productivity (GTFP) and analyze the influence mechanisms and heterogeneity among them. The results of the study are as follows: First, graduate student expansion and innovative human capital are the driving force for GTFP growth in China, and graduate student size expansion indirectly boosts GTFP by promoting the supply of innovative human capital, with a 73% mediating effect. Second, the effects of graduate student expansion and innovative human capital on GTFP show a trend of diminishing marginal benefits, and the estimated coefficient of the effect of graduate student scale share on GTFP has an inverted U-shaped relationship. Third, there is a significant threshold feature of industrial structure upgrading in the influence of graduate student scale expansion on GTFP. The study makes suggestions in terms of expanding of the graduate students, rationalizing the enrollment structure of universities, optimizing the regional industrial structure and creating a social innovation environment.

**Keywords:** graduate student expansion; innovative human capital; GTFP; threshold effect

## 1. Introduction

With the advancement of industrialization and urbanization, China's economy has achieved high growth rates, but accompanying environmental problems have become increasingly serious [1]. Some studies have shown that China's high economic growth is largely at the cost of environmental degradation, and that traditional uneven growth has caused serious ecological and environmental pressure, with resource and environmental problems threatening the lives of residents, as well as the green and sustainable development of the economy [2]. The economic competition of GDP tournaments and uneven economic growth has led to the emergence of problems such as wasteful production input and low efficiency of green production [3]. In this critical period of China's economic and social transformation and the rigid constraints of resources and environment, green sustainability will become an inevitable choice for China's future economic development. Meanwhile, achieving green transformational development and eco-innovation is an inevitable choice to realize green and sustainable economic development. In particular, improving green total factor productivity (GTFP) is a key aspect of eco-innovation [4]. Green total factor productivity (GTFP) is a comprehensive consideration of energy resources, ecological environment and other constraints to examine the efficiency and quality of the use of a country or region's input factors, reflecting the regional resource allocation capacity, environmental governance capacity and sustainable development capacity [5]. The Chinese government also attaches great importance to green total factor productivity (GTFP). In March 2021, the Chinese government issued a programmatic document to guide national development over the next 5 and 15 years—the Outline of the 14th Five-Year Plan

of the National Economic and Social Development of the People's Republic of China and Vision 2035—which proposed that within 5–15 years China needed to widely develop green production and lifestyle, a steady decrease in carbon emissions after reaching the peak, a more rational allocation of energy resources and a significant increase in utilization efficiency [6].

GTFP can be significantly enhanced by starting with human capital and enhancing the skill level of the workforce [7]. With the release of China's demographic dividends since the 21st century and the expansion of graduate students since 2010, higher-order human capital has shown a stronger driving force for the development of a green innovation economy [8]. Previous studies have focused on analyzing the relationship between factors such as regional economic level, industrial structure, market conditions and human capital and GTFP [9,10]. However, research involving graduate student expansion and GTFP is still relatively limited, and the mediating effect of innovative human capital on the relationship between graduate student expansion and GTFP remains to be explored. Surprisingly, China began a major expansion of higher education in 1999 and a major expansion of professional degree postgraduate students in 2010. In just 20 years, the gross enrollment rate of higher education in China rose from 17% in 2003 to 57.8% in 2021 [11]. The number of China's graduate students increased from 650,000 in 2003 to 3.14 million in 2020; an almost fivefold increase. The expansion of graduate students, on the one hand, may directly affect the efficiency of resource use and pollutant emissions by improving the production skills of the workforce [12], which involves a change in the production methods of individuals and the improvement of their management [13,14]. On the other hand, in the context of environmental pollution in China largely being influenced by the pollution emissions and backward technology of regional enterprises [15], green technology innovation is fundamental to achieve energy saving and emission reductions, improve quality and efficiency, and solve environmental pollution problems. Graduate student expansion enhances the supply of innovative talent [16] and changes the traditional industrial structure of the region, creating new green economic value [17]. Furthermore, innovative human capital in turn further influences the green technological innovation of labor market enterprises [18], which indirectly affects the efficiency of resource utilization and pollution emissions, providing the main support and driving force for enhancing GTFP [19].

Current research has examined the positive relationship between innovative human capital and GTFP [20,21]. A study measured the GTFP index for Chinese cities and verified the city size heterogeneity of GTFP development in China [22]. However, few studies have considered the relationship between graduate student expansion, innovative human capital and GTFP in the macro-context of developing countries. Hence, this paper intends to empirically test the relationship between graduate student expansion, innovative human capital and GTFP through a panel data model. Then, the mediating effect of innovative human capital will be analyzed using a mediation model, and the heterogeneity of the impact of graduate student expansion and innovative human capital on regions with different GTFP levels will be analyzed based on quantile regression. Finally, the threshold characteristics of graduate student expansion on GTFP at the level of industrial structure will be estimated to provide a reference framework for adjusting the human capital structure for regions at different industrial levels.

The innovative contributions of this study include, firstly, filling the research gap left the lack of studies that have explored the impact of education on GTFP by exploring the impact of education on national GTFP and its heterogeneity from the perspective of graduate education expansion. Secondly, the existing literature focuses on the non-linear relationship between industrialization [23] and innovation [24] on GTFP and the threshold effect of industrial structure, and has not explored the threshold effect of industrial structure in postgraduate expansion. This study considers the impact of the threshold effect of industrial structure under the expansion of postgraduate education, and provides a reference framework for adjusting the human capital structure in regions at different industrial levels. In conclusion, our study has implications for developing countries, especially those in the

process of green transformation development, in terms of public education policy design and industrial structure optimization.

## 2. Literature Review and Hypothesis

Some research on higher education and GTFP has focused on human capital and total factor productivity (TFP). The new economic growth theory proposed by Romer and Lucas argued that education could improve the quality of human capital and promote lasting economic growth, and that the higher the level of education (especially higher education), the more innovative human capital was formed, which in turn promoted technological progress and increased TFP [25]. Several empirical studies have also shown that higher education expansion significantly enhanced higher-order human capital accumulation and contributed to regional and national TFP [26,27]. Liu and Bi (2019) [28] further noted that higher education expansion had a positive and significant impact on TFP and thus on green economy sustainability. In addition, some studies have focused on higher education and green economic development, such as Lee and Heijden (2019) [29], Gao et al. (2020) [30] and Golbazi et al. (2020) [31], whose studies all reflect the positive link between higher education and green economic development, and consider higher education as an important green and innovative economic engine in the local economy, which has a positive impact on regional green economic development, green city building and green facility perceptions.

However, few studies directly reflect the impact of higher education expansion on GTFP. Wang & Wu (2021) [32] studied the differential effects of different levels of educational human capital on GTFP, with higher education human capital significantly contributing to regional GTFP, while primary education human capital suppressed GTFP. Some research has focused on the impact of innovative human capital and innovation factors on GTFP, such as Xiao and You (2021) [33], who presented the relationship between innovative human capital and GTFP and found that innovative human capital enhanced the level of regional innovation and achieved efficiency growth with less labor input so that GTFP could be enhanced. Additionally, Liu et al. (2020) [24] found that innovation in China had a significant positive effect on GTFP, and this effect was more pronounced in regions where human capital was concentrated. Zhao and Xu (2022) [34] also indicated the importance of human capital and innovation and advocated that the state needed to increase public spending to promote a green economy by enhancing human capital and technological innovation. Jin et al. (2022) [21] similarly present the positive impact of innovative human capital on GTFP while being subject to environmental regulation. Based on the above review, this study proposes the following research hypotheses:

**H1.** *Graduate student expansion has a positive impact on GTFP.*

**H2.** *Innovative human capital plays a mediating role in the relationship between graduate student expansion and GTFP.*

**H3.** *There is heterogeneity in the impact of graduate student expansion and innovative human capital on regions with different GTFP quantile levels.*

**H4.** *There is a threshold effect of industry structure on the impact of graduate student expansion on GTFP.*

## 3. Materials and Methods

### 3.1. Data Sources

This study examines the relationship between graduate student expansion, innovative human capital and GTFP using panel data for 30 Chinese provinces and cities (excluding Taiwan, Hong Kong and Inner Mongolia) from 2005 to 2018, with expansion mainly referring to the increase in the level of higher education enrollment in a region (province/municipality), which was obtained from the official website of the Chinese Ministry of Education. The data on innovative human capital were calculated based on the "China Labor Statistical Yearbook", "China Science and Technology Statistical Yearbook"

and "China Human Capital and Labor Economy Research Center" for each year. The data of GTFP were measured using the SBM-DEA model, and the original data were obtained from the "China Statistical Yearbook", "China Energy Statistical Yearbook" and "China Environmental Statistical Yearbook". The data of other control variables and threshold variables were obtained from the official website of the National Bureau of Statistics.

*3.2. Variables*

GTFP evaluation: The outcome variable of the study is green total factor productivity (GTFP). GTFP is based on the traditional total factor productivity (TFP) indicator and incorporates environmental pollution indicators into the framework of the accounting system as an endogenous variable of economic growth, which is an important basis to judge whether the region can achieve long-term sustainable development. There are many ways to calculate GTFP, including production function method, growth accounting, SFA, DEA, SBM-DEA and so on. In the face of GTFP calculations with multiple factor inputs and outputs, we referred to the SBM-DEA (slack-based measure–data envelopment analysis) measure created by Chung et al. (1997) [35]. This model, as a non-parametric efficiency analysis method, has advantages that other methods cannot match, such as not fixing data units, not specifying specific forms of production functions, etc. It also integrates the relationship between multiple factor inputs, desired output (GDP) and non-desired output (environmental pollution indicators), through which the GTFP index of each province can be obtained, and a higher index indicates that the region can have a higher economic growth rate with certain resource inputs while reducing environmental pollution. Based on the GTFP input and output indicator system established by Xie et al. (2021) [36], the input indicators of GTFP include capital, energy and labor, where capital input is replaced by capital stock, and the physical capital stock is estimated using the perpetual inventory method [37]. Energy inputs are selected from the total energy consumption of each province and city [38], and labor inputs are selected from the total number of people employed in industries in each province over the years. At the same time, the output indicators of GTFP include desired output and non-desired output, where the desired output is selected as GDP value, which contains nominal GDP and real GDP. Nominal GDP is inflated with inflation due to the annual GDP, so the CPI index is used to deflate GDP to obtain real GDP in the research process. On the issue of accounting for non-desired outputs, established studies have shown the general need to consider atmospheric pollution carbon dioxide ($CO_2$) or sulfur dioxide ($SO_2$) emissions [39] and industrial wastewater emissions [40], which were also selected for inclusion in non-desired outputs in this study. Based on the mentioned input and output indicators, we constructed production possibility sets of desired and non-desired outputs, and measured the GTFP values of 30 Chinese provinces and cities using the SBM-DEA model. The model assumes the existence of J DMUs (sets of units); each DMU corresponds to N inputs ($X_n$), while the corresponding desired outputs are I $Y_i$ and k non-desired outputs $b_k$, and the following super-efficient SBM model is constructed:

$$Sv = \min_{\lambda, \chi, y, b} \frac{1 - \frac{1}{N}\sum_{n=1}^{N}\frac{X_{n0}}{X_{m0}}}{1 + \frac{1}{I+k}\left(\sum_{i=1}^{I}\frac{y_i}{y_{i0}} + \sum_{k=1}^{k}\frac{y_k}{y_{k0}}\right)} \tag{1}$$

$$s.t. \sum_{j=1}^{j}\lambda_j Xnj \leq Xn \quad \sum_{j=1}^{j}\lambda_j X_{kj} \geq yi \quad \sum_{j=1}^{j}\lambda_j X_{kj} = b_k \quad \sum_{j=1}^{j}\lambda_j = 1) \tag{2}$$

Graduate Student Expansion: The expansion of graduate students refers to an increase in the number of graduate students enrolled; the variable used was the number of graduate students enrolled in each province, and the value was logarithmically included in the model. In addition, the ratio of the number of graduate students enrolled to the total number of students enrolled was selected to reflect whether regional universities mainly

train graduate or undergraduate students, and this variable was expressed in the form of a percentage, which can be directly included in the model analysis.

Innovative Human Capital: Innovative human capital is derived from the concept of human capital, and scholars have carried out studies on the definition of innovative human capital. For example, Kataoka (2017) [41] proposed that innovative human capital is a reorganization of human resources and production factors, and unlike human capital, which is measured by a single indicator (e.g., average years of education), innovative human capital reflects more of a combination of competencies. McGuirk et al. (2015) [42] added other characteristics of individuals, such as job satisfaction, to the traditional measure of human capital. Studies have been conducted to measure regional innovative human capital, usually using years of education in the workforce, R&D personnel or R&D financial input, lifetime earnings approach, etc. [32,43,44]. For a more scientific characterization of regional innovative human capital, we combined these elements—i.e., we selected the data obtained from the three measures of years of education in the labor force, R&D full-time equivalent and J-F income—and added and took logarithms after standardization and de-quantization. The original standardized data were values between 0 and 1, and there were negative values after taking logarithms, but the region's innovative human capital should not be taken as negative.

Control variables: There are many factors affecting GTFP, and in addition to education and human capital factors, existing studies tend to select urban industry, urbanization process, economic level, population size and market regulation conditions as relevant control conditions [45–48]. This study selected industrial structure (the ratio of tertiary industry value added to secondary industry value), urbanization rate, GDP per capita, population size and marketization index as control variables.

Threshold variables: The relationship between graduate student size and GTFP is not necessarily linear and may also be influenced by the threshold effect of industrial structure. Postgraduate education is the driving force in the green technology innovation system, and the expansion of postgraduate education may be able to facilitate the restructuring of industries from the lowest to the higher levels [30]. The increase in the proportion of tertiary industry is conducive to improving the efficiency of resource use, allowing high-level human capital, such as graduate students, to better create a green economy and improve green total factor productivity [49]. Based on this, this paper infers that the impact of the size of postgraduate education on GTFP may be influenced by the structure of the industry. Accordingly, we chose industrial structure as the threshold variable.

### 3.3. Data Analysis

Panel Regression Model: We constructed the following panel data model, where $Ln(GTFP)_{it}$ denotes the logarithm of the region's GTFP level, $Ln(GS)_{it}$ denotes the logarithm of graduate student size, $PGS_{it}$ indicates the graduate student proportion, and $Ln(IHC)_{it}$ denotes the regional innovative human capital. In addition, industrial structure, urbanization rate, GDP per capita, population size, and marketization index were included in the econometric model as control variables that may have an impact on GTFP ($ConVar_{it}$), with $\alpha_i$ as the model intercept and $\varepsilon_{it}$ as the residual term.

$$Ln(GTFP)_{it} = \alpha_i + \beta_1 Ln(GS_{it}) + \beta_2 PGS_{it} + \beta_3 ConVar_{it} + \varepsilon_{it} \tag{3}$$

$$Ln(GTFP)_{it} = \alpha_i + \beta_1 Ln(GS_{it}) + \beta_2 PGS_{it} + \beta_3 Ln(IHC)_{it} + \beta_4 ConVar_{it} + \varepsilon_{it} \tag{4}$$

Mediation effect model: We used a mediating effect model to determine whether the mediating effect of innovative human capital between graduate student expansion and GTFP held true, and to analyze the share of the degree of mediation. The bias-corrected nonparametric percentile bootstrap estimation method was used to calculate the mediating effect estimates by randomly replicating the sample (N = 420) 5000 times and to estimate the 95% confidence interval of the mediating effect.

Quantile regression: To estimate the heterogeneity of the effect of graduate student size on areas that are at different GTFP levels, we used quantile regression models for estimation. The advantage of quantile regression is to more comprehensively describe the marginal impact of the independent variable on the conditional quantile of the dependent variable at different quartiles, so as to adopt precise and differentiated supportive strategies to promote GTFP in order to optimize the marginal benefits of inputs and outputs. The quantile regression equation is expressed as follows, where q denotes different quantile points, and y and x represent the dependent and independent variables, respectively:

$$Q_\theta(y|X) = Min\beta_q \sum_{i:yi \geq xi\beta q}^{n} q|y_i - x_i\beta_q| + \sum^{n} i : y_i \leq x_i\beta q(1-q)|y_i - x_i\beta_q| \quad (5)$$

Threshold effect model: This study focuses on analyzing the threshold effect of industrial structure when analyzing the impact of graduate student expansion on GTFP; for promoting GTFP, there may be a marginal benefit optimum (i.e., threshold) for regional industrial structure. The model formula for the threshold regression is as follows, where $Y_i$ is the dependent variable GTFP, $X_{1i}$ is the core explanatory variable affected by the threshold—in this case, graduate student size, $X_{2i}$ is the non-core explanatory variable not affected by the threshold-influenced non-core explanatory variables, $\varphi$ is the true threshold value estimated by proxy, and $qi$ and $D(X)$ represent the threshold variable and the indicative function, respectively. When $qi \leq \varphi$, then $D(qi) = 1$, and vice versa $D(qi) = 0$. If a threshold does exist for industrial structure, it indicates that there is a nonlinear effect of graduate student expansion on promoting GTFP, and a different trend of the effect before and after the threshold.

$$Y_i = \alpha_i + \beta'_{11}X_1 D(qi \leq \varphi) + \beta'_{12}X_{1i}D(qi > \varphi) + \beta'_2 X_{2i} + \varepsilon_i = \alpha_i + \theta X_i(\varphi) + \varepsilon_i \quad (6)$$

## 4. Results

### 4.1. Descriptive Statistics and Correlation Test

Descriptive statistics for the variables are presented in Table 1 below, which presents the sample size, mean, standard deviation, minimum and maximum values for each variable. It should be noted that in order to better explain the relationship between the model variables, reduce the absolute values of the variables and eliminate possible heteroskedasticity, the variables with large dispersion, such as GTFP, innovative human capital, graduate scale, GDP per capita and population size, were treated as logarithms. Meanwhile, both the independent and dependent variables were taken as logarithms. The economic meaning of the coefficients of the independent variables estimated by the model is "elasticity" when both the independent and dependent variables are logarithmic. "Elasticity" means that when the independent variable increases by 1%, the dependent variable changes by $100 \times \beta$% on average.

**Table 1.** Descriptive statistics of each variable.

| Variables | Obs | Mean | Std. Dev. | Min | Max |
|---|---|---|---|---|---|
| GTFP | 420 | 0.258 | 0.473 | −1.182 | 1.520 |
| Innovative human capital | 420 | −0.181 | 0.363 | −1.022 | 0.952 |
| Graduate scale | 420 | 10.454 | 1.128 | 6.796 | 12.905 |
| The proportion of graduate students | 420 | 0.067 | 0.057 | 0.015 | 0.404 |
| Industrial structure | 420 | 1.146 | 0.628 | 0.527 | 5.022 |
| Urbanization rate | 420 | 53.534 | 13.892 | 26.870 | 89.600 |
| GDP per capita | 420 | 10.418 | 0.647 | 8.528 | 11.851 |
| Population size | 420 | 8.176 | 0.746 | 6.297 | 9.337 |
| Market index | 420 | 0.653 | 0.191 | 0.233 | 1.171 |

The regional distribution of growth and changes in GTFP and graduate student size is shown in Figure 1. It can be seen that all regions in China achieved some degree of growth in GTFP during 2005–2018. However, the growth and changes varied greatly, with

Beijing, Tianjin, Shanghai, Guangdong, and Jiangsu in the eastern region performing better in GTFP, while most provinces in the western and northeastern regions performing poorly in GTFP. The comparison reveals that there are some similarities between the growth of graduate student scale and the growth of GTFP, and that regions demonstrating fast growth of graduate student scale may also achieve a large degree of GTFP growth.

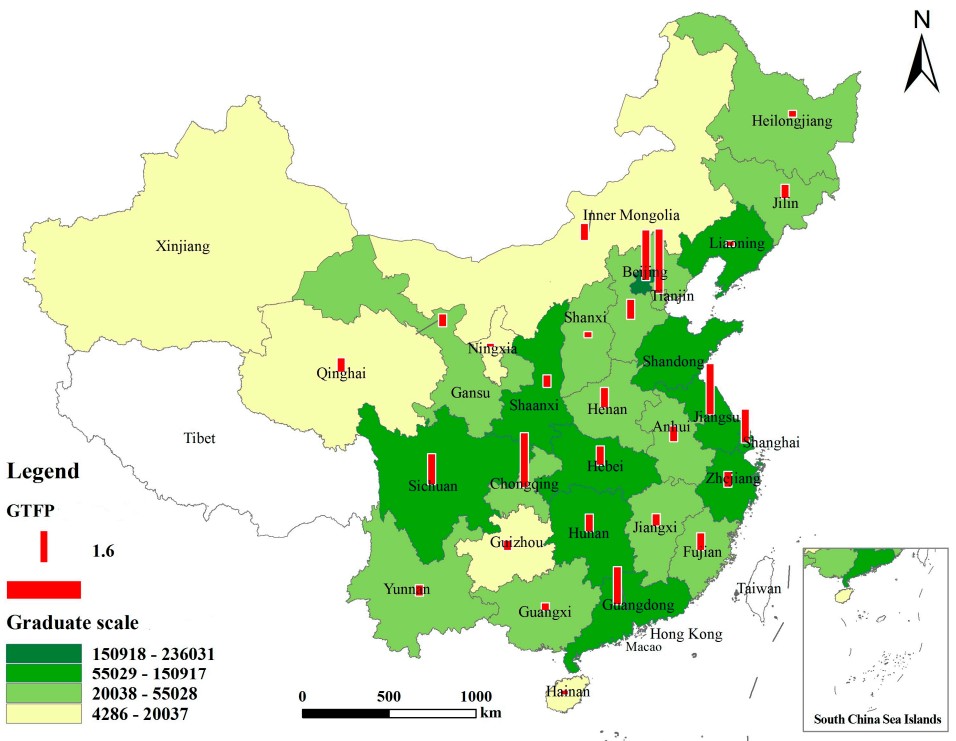

**Figure 1.** Graduate scale growth and GTFP growth.

Figure 2 presents the scatter plot between graduate student size, innovative human capital and GTFP. It can be seen that graduate student size was positively associated with innovative human capital and GTFP, respectively. In addition, the scatter trend between graduate student size and innovative human capital appeared to be more concentrated with a higher slope magnitude. The presentation of scatter plots helps us understand whether there is a positive association between these three variables, and lays the foundation for the subsequent model analysis.

*4.2. Panel Regression Model*

We used the panel unit root test and panel cointegration analysis before modeling the panel data. The panel unit root test refers to the inclusion of each cross-section of the variables in the panel data as a series to test whether the panel data are a smooth time series. Using the LLC unit root test, it was found that the original series of all variables rejected the original hypothesis of "existence of unit root" at the significance level of $p < 0.01$. Therefore, we concluded that the variables in the panel data were relatively smooth overall, and could further determine whether there was a long-term equilibrium relationship between these variables. The Kao test was used to analyze the panel cointegration, and it was found that the original hypothesis of "no cointegration" was rejected at the $p < 0.001$ level of significance for all the test indicators, which supported the existence of a long-term equilibrium relationship in the panel data and the subsequent use of the panel regression model.

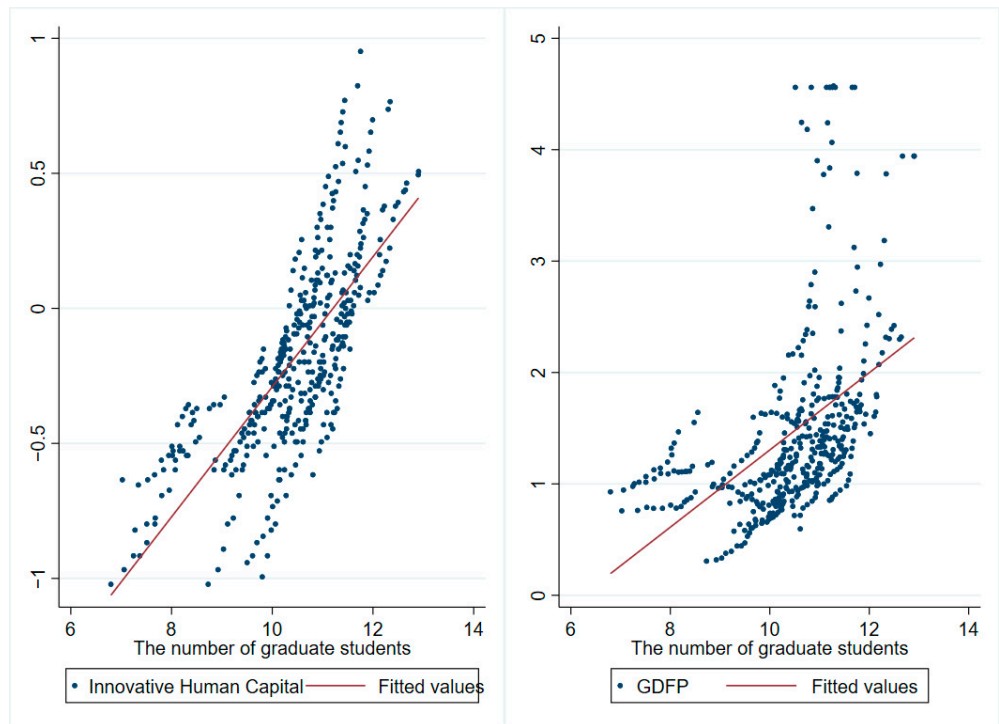

**Figure 2.** Scatter plot of graduate student size, human capital and GTFP.

The Hausman test was used to determine whether to choose the random effects model or the fixed effects model, and the test results revealed (Table 2) that there was no significant difference between the fixed effects and random effects in the coefficient estimates, and therefore the (FE-RE) difference coefficient was not significant. Accordingly, the random effects model was considered to be more efficient than the fixed effects model and the random effects model (Re) was selected.

**Table 2.** Hausman test results of fixed effect.

| Variable | (b) FE | (B) RE | (b-B) Difference | Sqrt S.E. |
|---|---|---|---|---|
| Innovative human capital | 0.257 | 0.280 | −0.023 | 0.015 |
| Graduate scale | 0.127 | 0.102 | −0.050 | 0.049 |
| The proportion of graduate students | 3.515 | 2.888 | 0.627 | 0.271 |
| Industrial structure | 0.060 | 0.055 | 0.005 | 0.012 |
| Urbanization rate | 0.009 | 0.005 | 0.004 | 0.002 |
| GDP per capita | 0.133 | 0.155 | −0.023 | 0.015 |
| Population size | −0.009 | 0.132 | −0.142 | 0.125 |
| Market index | 0.519 | 0.544 | −0.025 | 0.013 |
| LR $\chi^2$ | | | 8.71 | |
| Prob > $\chi^2$ | | | 0.367 | |

Based on the panel regression model, the effect of graduate student expansion on GTFP was obtained, and the results are shown in Table 3. Model 1 is the model without the inclusion of graduate student size and graduate student ratio variables; it can be seen that with the inclusion of relevant control variables, innovative human capital had a positive and significant effect on GTFP ($p < 0.01$) with an elasticity coefficient of 0.335, indicating that for every 1% increase in regional innovative human capital level, the regional GTFP was able to increase by 0.335%. In the control variables, industrial structure, GDP per capita and marketability index all had a significant positive effect on GTFP. Additionally, Model 2 incorporates the variables of regional graduate student size and graduate student

ratio, in which innovative human capital still had a positive and significant effect on GTFP ($p < 0.01$), but the elasticity coefficient decreased from 0.335 to 0.280, while graduate scale and graduate student ratio both had a positive and significant effect on GTFP ($p < 0.05$). The elasticity output coefficient of graduate scale on GTFP was 0.102 and the coefficient of graduate student ratio on GTFP was 2.888, which indicates that the expansion of graduate student size promotes the level of GTFP.

**Table 3.** The impact of postgraduate scale expansion on GTFP.

| Variable | Model 1 | | | | Model 2 | | | |
|---|---|---|---|---|---|---|---|---|
| | Coef. | S.E. | t | $p$ | Coef. | S.E. | t | $p$ |
| Graduate scale | | | | | 0.102 | 0.048 | 2.125 | 0.016 |
| The proportion of graduate students | | | | | 2.888 | 0.576 | 5.014 | 0.000 |
| Innovative human capital | 0.335 | 0.084 | 3.988 | 0.000 | 0.280 | 0.083 | 3.373 | 0.001 |
| Industrial structure | 0.160 | 0.028 | 5.714 | 0.000 | 0.055 | 0.035 | 1.571 | 0.116 |
| Urbanization rate | −0.001 | 0.003 | −0.333 | 0.715 | 0.005 | 0.004 | 1.250 | 0.204 |
| GDP per capita | 0.163 | 0.041 | 3.976 | 0.000 | 0.155 | 0.042 | 3.690 | 0.000 |
| Population size | 0.084 | 0.082 | 1.024 | 0.306 | 0.132 | 0.098 | 1.347 | 0.176 |
| Market index | 0.598 | 0.076 | 7.868 | 0.000 | 0.544 | 0.075 | 7.253 | 0.000 |
| intercept | −2.570 | 0.790 | −3.253 | 0.001 | −2.462 | 0.805 | −3.058 | 0.002 |
| model form | RE | | | | RE | | | |
| Within-$R^2$ | 0.742 | | | | 0.760 | | | |
| Between-$R^2$ | 0.372 | | | | 0.354 | | | |
| Wald $\chi^2$ | 1132.84 *** | | | | 1229.55 *** | | | |

Note: *** $p < 0.01$.

*4.3. Mechanism Analysis*

Figure 3 shows the results of the path coefficients of the impact of graduate student expansion on regional GTFP. As shown in Figure 3, the model jointly explained 42.4% of the variance of GTFP, graduate student expansion explained 56.0% of the variance in innovation human capital, and the path of graduate student expansion affecting GTFP contains two parts: one is that the increase in graduate student size can directly affect GTFP ($\beta = 0.128$, $p < 0.01$), and the other is by promoting the increase in innovative human capital ($\beta = 0.748$, $p < 0.01$), which then has a positive and significant effect on GTFP ($\beta = 0.549$, $p < 0.01$).

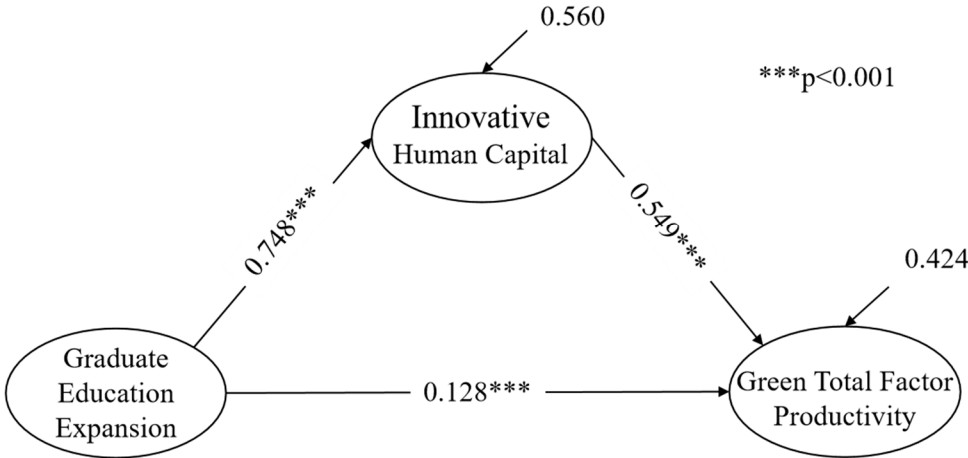

**Figure 3.** Mediation effect model diagram.

Bootstrap estimation was used to analyze the specific mediating effect of innovative human capital (Table 4). The results show that the mediating effect of innovative human capital holds ($\beta = 0.411$, $Z = 10.024$), and the proportion of the mediating effect was

calculated to be 76.3%, which is a strong mediating effect. Thus, the graduate student expansion largely enhances the regional GTFP by promoting innovative human capital first.

**Table 4.** Test of the mediation effect of innovative human capital.

| Mediation Test | Point Estimate | Product of Coefficients | | Bootstrapping | | | | Effect Ratio |
|---|---|---|---|---|---|---|---|---|
| | | | | Bias-Corrected 95% CI | | Percentile 95% CI | | |
| | | SE | Z | Lower | Upper | Lower | Upper | |
| Total | 0.539 | 0.024 | 22.458 | 0.485 | 0.586 | 0.492 | 0.587 | - |
| Direct | 0.128 | 0.046 | 2.783 | 0.039 | 0.224 | 0.044 | 0.227 | 23.7% |
| Indirect | 0.411 | 0.041 | 10.024 | 0.326 | 0.486 | 0.324 | 0.483 | 76.3% |

Note: 5000 bootstrap samples.

### 4.4. Heterogeneity Test

To test whether there is heterogeneity in the effects of graduate student expansion and innovative human capital on regions with different GTFP quantile levels, a conditional quantile regression model was applied to estimate the regression results for the five quantile points of 0.1, 0.3, 0.5, 0.7, and 0.9. Then, the estimated coefficients of the five quantile points were connected to form a trend graph (Figure 4), reflecting the difference in the change in benefits brought to regions with different GTFP levels. In the standard quantile regression model QR, graduate student expansion and innovative human capital had a significant positive impact on the increase of GTFP regardless of the GTFP level. However, for areas where GTFP was in the 60th percentile, the effect of postgraduate expansion on the increase in GTFP became negative and insignificant. Meanwhile, the marginal benefits of overall graduate student expansion and innovative human capital show a decreasing trend. This result also indicates that boosting graduate student size and promoting innovative human capital are more marginal benefits for regions with lower GTFP levels and can help green-economy-disadvantaged regions to rapidly improve their GTFP.

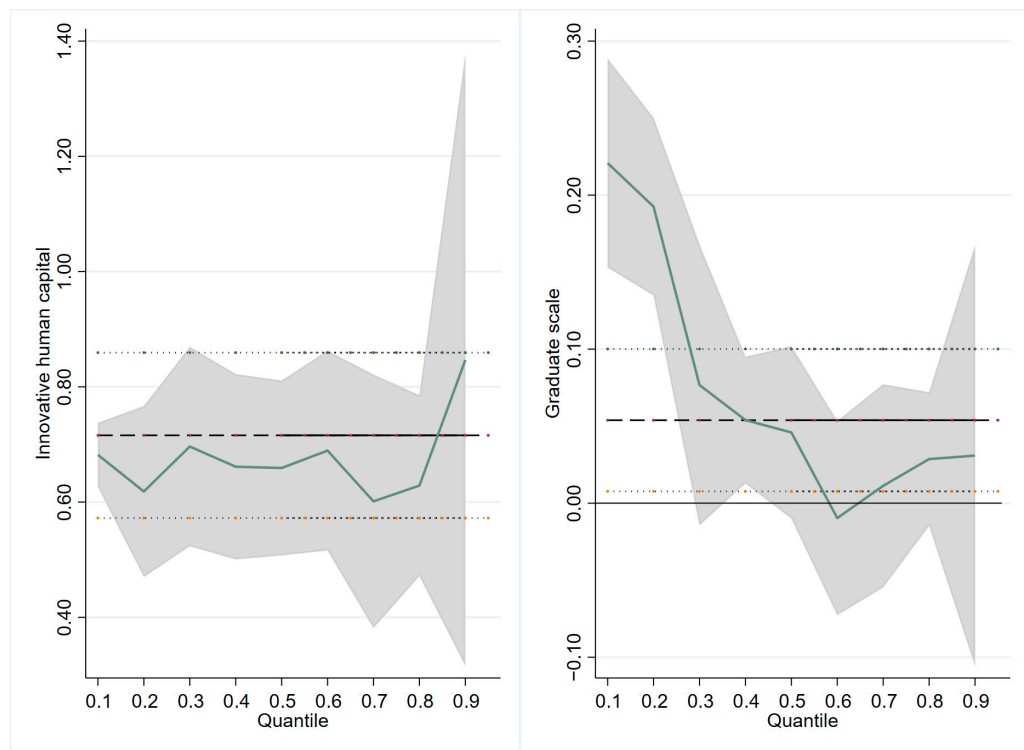

**Figure 4.** Conditional quantile regression.

### 4.5. Threshold Effect Test of Industrial Structure

Threshold regression was used to analyze whether there is a threshold effect of industrial structure on the promotion of GTFP by graduate student expansion, so as to more efficiently benchmark the marginal optimum. Figure 5 presents the graph of the LR test under the double threshold confidence interval, where the horizontal dashed line is the 95% confidence level and the curve is the line connecting the search points of each threshold, from which it can be seen that there are two prominent values of obvious industrial structure, indicating that the double threshold regression estimation was more accurate.

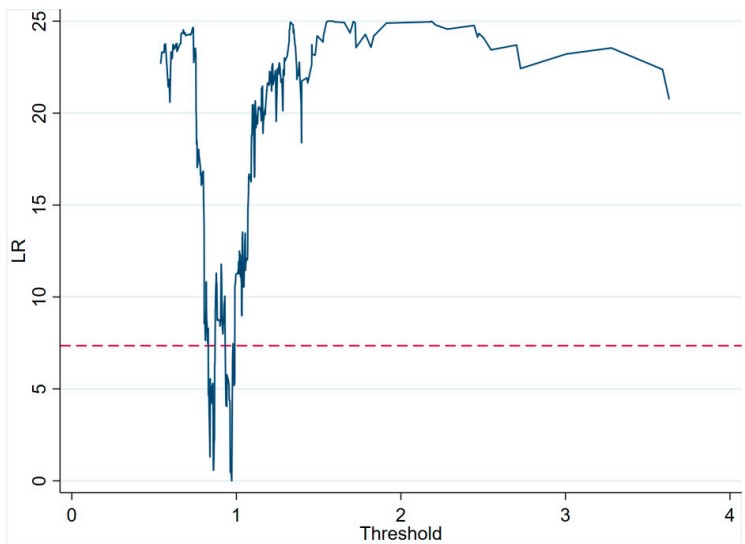

**Figure 5.** Threshold LR test.

The results of the double-threshold regression estimation of industrial structure are shown in Table 5; when the industrial structure index is below 1.387, the relationship between graduate student expansion and GTFP is not significant ($\beta = 0.082$, $p > 0.1$). When the industrial structure index is in the range of 1.387–3.006, the coefficient of graduate student expansion is raised to 0.095 and passes the 1% level significance test. When industrial structure index reaches the critical value of 3.006, the elastic output to GTFP is raised to 0.122 again ($p < 0.05$); that is, when the industrial structure index is greater than 3.006, the highest benefit to human capital accumulation in the region. In conclusion, the industrial structure follows the principle of threshold restriction; that is, the industrial structure needs to reach a certain threshold standard for graduate students and innovative human capital to promote regional GTFP more efficiently. The optimization of industrial structure provides the necessary conditions for human capital and industrial coordination. In particular, the further development of the tertiary industry lays a good industrial foundation and motivation for innovative human capital and accelerates the development of GTFP. Observing the industrial structure index of 30 Chinese provinces and cities in 2018, 17 regions had an industrial structure index below 1.387, 12 regions had an industrial structure index in the range of 1.387–3.006, and only Beijing had an index higher than 3.006, which indicates that the current industrial structure of most regions is not sufficient to meet the optimal environment for GTFP enhancement.

**Table 5.** Threshold effect test of industrial structure.

| Double Threshold | Coef. | Std. Err. | t | *p*-Value |
|---|---|---|---|---|
| $Q_i \leq \varphi_1$ | 0.082 | 0.050 | 1.63 | 0.104 |
| $\varphi_1 < Q_i \leq \varphi_2$ | 0.095 | 0.050 | 1.90 | 0.058 |
| $Q_i > \varphi_2$ | 0.122 | 0.051 | 2.38 | 0.018 |
| Control variable | YES | | | |
| Threshold/$\varphi$ | 1.387–3.006 | | | |
| Within-$R^2$ | 0.755 | | | |
| Between-$R^2$ | 0.370 | | | |
| F | 128.65 *** | | | |

Note: *** $p < 0.001$.

## 5. Conclusions and Policy Implications

### 5.1. Conclusions

Graduate student expansion provides innovative human capital to the labor market, which provides important support and a main driving force for green economic development, and is important for promoting GTFP. This paper found that graduate student expansion and innovative human capital were the driving force for GTFP growth in China, and that graduate student expansion could partly directly promote regional GTFP, but to a large extent indirectly enhanced GTFP by promoting the supply of innovative human capital, with a mediating effect of about 73%. In addition, the effects of graduate student expansion and innovative human capital on GTFP showed a trend of diminishing marginal benefits, and the estimated coefficient of the effect of graduate student scale share on GTFP had an inverted U-shaped relationship. There was a significant threshold feature of industrial structure upgrading in the impact of graduate student expansion on GTFP, and the green effect of graduate student expansion gradually increased with the growth of tertiary industry.

### 5.2. Policy Implications

Our findings provide evidence and insights for developing countries to enhance GTFP from the perspective of education and innovative human capital. The study recommends, first, that the education sector needs to further expand the size of graduate schools and rationalize the enrollment structure of universities. The current regional differences in China's GTFP are too large, and the more economically backward regions have lower levels of GTFP, while Chinese graduate training universities are concentrated in economically developed regions. There is a similar situation in developing countries as well. Therefore, we suggest that steps should be taken to optimize the spatial allocation structure of graduate enrollment sites, reasonably plan and arrange the number and types of higher education institutions, focus on strengthening tilting efforts to the central and western regions, and pay attention to the supply of graduate students and innovation capacity enhancement in the disadvantaged areas of GTFP development. Second, provincial departments need to promote industrial restructuring and sustainable improvement at the industrial level. Since most Chinese provinces have not yet reached the optimal industrial structure index for promoting GTFP, provinces and regions with developed green economies need to further increase support for green industries, play the role of talents and innovative technologies for economic growth, and accelerate the economic transformation rate of technologies. In particular, in regions with weak green economies, the level of industrialization and the penetration of green technologies are still not high, so it is more necessary to pay attention to the central role of innovative human capital in industrial restructuring, and reduce the damage to natural resources in the process of industrial change. Third, the government is supposed to correctly and effectively guide social enterprises to invest in innovation projects and encourage them to increase R&D investment through formal and structured tax incentives. At the same time, increasing the establishment of innovative R&D

institutions and providing a better environment for graduate students to conduct research and innovation to promote the upgrading of green industries in the region are necessary.

### 5.3. Research Limitations

This study enriches the research related to education on GTFP. However, due to the limitations of personal ability and objective conditions, this study has the following shortcomings:

First, the study took Chinese provincial-level data as the sample and failed to obtain a sample of Chinese cities as the unit of data, so the coefficient estimates obtained in the panel data model analysis may produce some bias and the model estimation results could be more accurate if more detailed data of a large sample of cities were available.

Second, the independent variable of this study, graduate student expansion, is not detailed enough and lacks data on different disciplines (e.g., science and technology, humanities majors), etc. In a follow-up study, more innovative and valuable conclusions could be obtained if data on different disciplinary types of graduate students can be collected in the future.

Finally, the internal mechanism of the impact of graduate student expansion on GTFP is complex. This study analyzes from the perspective of innovative human capital, but there may be other more complex mediating factors on the impact of graduate student expansion on GTFP. Future studies can consider further factors, such as environmental regulation and graduate student employment.

**Author Contributions:** Conceptualization, H.Y. and X.G.; methodology, H.Y.; software, X.G.; validation, X.G. and Q.Y.; formal analysis, X.G. and Q.Y.; investigation, H.Y. and X.G.; resources, X.G. and Q.Y.; data curation, H.Y. and X.G.; writing—original draft preparation, H.Y. and X.G.; writing—review and editing, Q.Y.; supervision, Q.Y.; project administration, X.G. and Q.Y.; funding acquisition, X.G. and Q.Y. All authors have read and agreed to the published version of the manuscript.

**Funding:** This research was funded by the National Education Sciences Planning of China, grant number BEA210114.

**Institutional Review Board Statement:** Not applicable.

**Informed Consent Statement:** Not applicable.

**Data Availability Statement:** The data presented in this study are available on request from the corresponding author. The data are not publicly available due to their usage in a production system.

**Conflicts of Interest:** The authors declare no conflict of interest.

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
