# Peer review of "Impact of Graduate Student Expansion and Innovative Human Capital on Green Total Factor Productivity"

_sustainability, doi:10.3390/su15021721_

Round 1
Reviewer 1 Report
- I think it might be helpful to elaborate more on the causal model being proposed here.
- Could the authors unpack the breakdown of graduate students -- presumably graduate students in STEM disciplines have a different impact on GTFP than graduate students in non-STEM disciplines (e.g., philosophy).
- How is "innovative human capital" being defined and operationalized?
- Could the authors unpack what an SBM-DEA model is to make this more accessible to readers? I do not think that acronym is even defined in the text.
Author Response
Thank you pretty much for your kind and valuable comments which contribute a lot to our article polishment. We have reviewed our articles according to the comments and explained as follows:
Point 1
I think it might be helpful to elaborate more on the causal model being proposed here.
Response 1
、Our study is based on a panel data model and does not have causal inferential models (e.g., RD, DID models) due to data limitations published by the Chinese government. But we control for the corresponding variables of economic, demographic, urban, and market factors, and also make analyses such as threshold regressions and heterogeneity, which we believe can largely corroborate the robustness of the results. At the same time, we looked for related studies that also did not make corresponding causal inference models, such as:
Wang, Yakun, Jingli Jiang, Dongqing Wang, and Xinshang You. 2023. "Can Mechanization Promote Green Agricultural Production? An Empirical Analysis of Maize Production in China" Sustainability 15, no. 1: 1. https://doi.org/10.3390/su15010001
Chen, Kai, Feng Guo, and Shuang Xu. 2022. "The Impact of Digital Economy Agglomeration on Regional Green Total Factor Productivity Disparity: Evidence from 285 Cities in China" Sustainability 14, no. 22: 14676. https://doi.org/10.3390/su142214676
Point 2
Could the authors unpack the breakdown of graduate students -- presumably graduate students in STEM disciplines have a different impact on GTFP than graduate students in non-STEM disciplines (e.g., philosophy).
Response 2
Our study is modeled based on panel data for 30 Chinese provinces and cities from 2005-2018. We also try to investigate the differences in the impact of graduate students in different disciplines (e.g., STEM and non-STEM), but unfortunately, we were unable to find the indicator in the data published by the Chinese government, and China does not have publicly available data on graduate students at the discipline level, such as STEM and non-STEM. In addition, the focus of our study is to explore the impact of graduate student size expansion on GTFP in general, and not specifically on the impact of disciplines, although we would be happy to explore this further if corresponding data were available.
Point 3
How is "innovative human capital" being defined and operationalized?
Response 3
Thank you for your kind comments. We found that due to the formatting of the article, the definition of the variable expression was omitted. We have added the definition and operationalization of "innovative human capital" in the paper.
Point 4
Could the authors unpack what an SBM-DEA model is to make this more accessible to readers? I do not think that acronym is even defined in the text.
Response 4
We have added some information about the SBM-DEA model.
Reviewer 2 Report
The author/s has/have done an excellent job, of providing a narrative which critical and important in the aspect of sustainability. The critical issue/s with the paper is the innovative human capital
what is the definition of innovative human capital?
How did the author/s measure innovative human capital?
The literature on innovative human capital is limited and what are the variables used?
The authors can use - McGuirk et al. (2015) measuring the impact of innovative human capital on small firms' propensity to innovate. Research Policy, 44(4), 965-976.
The author/s should include research limitations.
Author Response
Thank you pretty much for your kind and valuable comments which contribute a lot to our article polishment. We have reviewed our articles according to the comments and explained as follows:
Point 1
what is the definition of innovative human capital? How did the author/s measure innovative human capital? The literature on innovative human capital is limited and what are the variables used? The authors can use - McGuirk et al. (2015) measuring the impact of innovative human capital on small firms' propensity to innovate. Research Policy, 44(4), 965-976.
Response 1
We have supplemented the definition and operationalization of innovative human capital with the corresponding literature.
Point 2
The author/s should include research limitations.
Response 2
We have added research limitations.